# Experimental Study at the Reservoir Head of Run-Of-River Hydropower Plants in Gravel Bed Rivers. Part I: Delta Formation at Operation Level

**Christine Sindelar \*, Thomas Gold, Kevin Reiterer, Christoph Hauer and Helmut Habersack**

CD-Laboratory for Sediment Research and Management,
Institute for Hydraulic Engineering and River Research, Department for Water–Atmosphere–Environment,
BOKU-University of Natural Resources and Life Sciences Vienna, Muthgasse 18, 1190 Vienna, Austria;
thomas.gold@boku.ac.at (T.G.); kevin.reiterer@boku.ac.at (K.R.); christoph.hauer@boku.ac.at (C.H.);
helmut.habersack@boku.ac.at (H.H.)

**\*** Correspondence: Christine.sindelar@boku.ac.at

**Abstract:** This study concerns scaled physical model tests of the delta formation process at the head of a run-of-river hydropower plant (RoR). It forms part of a larger research project to provide a scientific base for RoR sediment management strategies in medium-sized gravel bed rivers. The physical model consisted of an idealized river having a width of 20 m, a mean slope of 0.005, a mean flow rate of 22 m³/s and a 1-year flood flow of 104 m³/s. The model scale was 1:20. For the experiments, five different grain sizes were used, covering a range of 14 to 120 mm at 1:1 scale. Experiments were carried out under mobile-bed conditions at flow rates which correspond to 50%–80% of a 1-year flood $HQ_1$. Even at the head of the reservoir, which is least influenced by the backwater effect of the RoR, sediment transport practically ceases for sediment fractions > 14 mm for a flow rate of $0.7 \times HQ_1$. The whole sediment load coming from the undisturbed upstream section accumulates at the head of the reservoir. This delta formation is accompanied by a substantial rise in water levels. A spatio-temporal scheme of the delta formation was derived from the experiments. The study proved that the delta formation increases the flood risk at the head of the reservoir. Conversely, reservoir drawdowns at flood events of high probability may be a promising strategy to enhance sediment connectivity under the specified boundary conditions.

**Keywords:** sediment management strategies; run-of-river hydropower plants; sediment mixtures; delta formation; sediment continuity

## 1. Introduction

Human activities and climate change-related effects have disturbed natural sediment dynamics in riverine systems worldwide. Recently, a comprehensive global survey on the status of rivers revealed that dams, including run-of-river hydropower plants (RoR), significantly affect river connectivity [1]. Austria has approximately 2770 RoRs, 87% of which are located in small and medium-sized rivers [2]. Efforts need to be taken to restore sediment connectivity, which is key contributor to eco-system services in riverine systems. Measures to prevent reservoir sedimentation and to improve sediment connectivity at RoRs are possible at catchment scale, in the reservoir, at the dam structure or by adapting the operation of the RoR [3–5]. At RoRs, the flow decelerates from the head of the reservoir towards the dam, with, consequently, the largest water depths upstream of the

obstruction. Sediment dynamics in rivers are mainly related to high flows and the consequently higher shear forces. In general, the power house of an RoR is typically located in a bay either on the left or right river bank. Close to the power house (PH), the flow usually separates into the main flow through the turbines and a large eddy with vertical axis in front of the dam. During flood events, when the turbines are not in operation, the flow has to pass through the flood gates which have a total inner width comparable to the original river width $W_0$. The river width $W_d$ at the RoR, consisting of the power house and the flood gate(s), is typically much wider than $W_0$ (Figure 1). In this paper, the impounded section is termed the "reservoir". The operation where the reservoir is drawn down by opening the flood gate(s), such that a free surface flow develops, is termed "flushing".

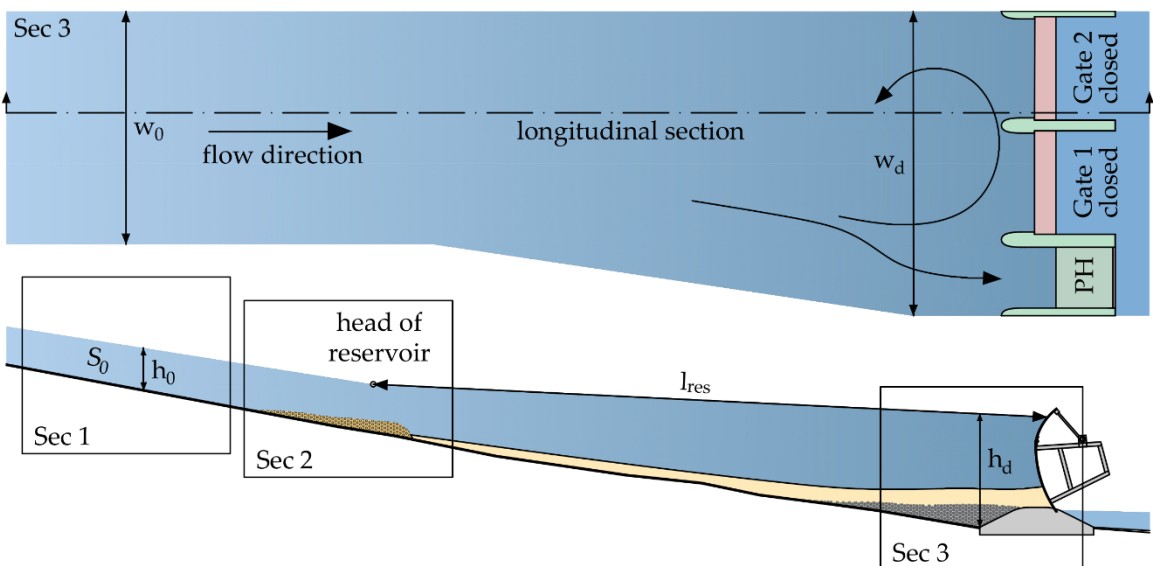

**Figure 1.** Scheme of an RoR hydro power plant; plan view of Sec 3 close to the dam (top) and longitudinal section (bottom).

RoRs have a small storage capacity, so depth and width variations are small compared to storage hydropower plants with huge reservoir volumes. Yet, the effect of varying water depths and river widths are dominating factors regarding sediment distribution in an RoR reservoir. Due to a heterogeneous sediment mixture, sorting effects of the incoming sediments may occur in longitudinal, lateral and vertical directions during turbine operation. Longitudinal sediment sorting in a 1500 m long reservoir of an RoR plant in the Mur River in Austria was reported by [6]. The collected sediment samples revealed a grain diameter decrease ($d_m$ and $d_{90}$) by a factor of about $10^{-2}$ from the head of the reservoir to the dam. At the head of an RoR reservoir, the coarsest fractions of the transported bed load material settle. Along the longitudinal axis of the reservoir smaller and smaller sediment particles start to be deposited until even suspended fractions settle.

In (gravel bed) rivers, the largest amount of sediments is transported as suspended load. Field surveys in gravel bed rivers report that suspended load accounts for up to 90% of the total sediment load [7]. In medium-sized gravel bed rivers, RoRs have hydraulic heads which are typically <10 m. They are classified as "low-head" hydropower. Their reservoirs are small with respect to length, width, depth and the ratio of reservoir storage capacity and mean annual runoff. Therefore, fine sediments do not contribute much to reservoir sedimentation. Most of the fine sediments pass the turbines or the flood gates as suspended load. In these rivers, the gravel fractions are essential to maintain important ecosystem functions, such as providing habitats and spawning gravel for fish [8,9]. Due to the impoundment, often these important sediment fractions are deposited at the head of RoR reservoirs, leading to delta formation, and are missing further downstream [3,6,10]. In the sense of sustainable sediment management strategies, countermeasures to prevent successive reservoir sedimentation are sought.

There is an increasing awareness among HPP operators that sediment management strategies are an integral part of the operation of a hydropower station or a chain of stations. The operation of an RoR must consider both economic and ecological aspects. Different legal frameworks, such asthe Water Framework Directive (WFD) [11] and the Revised Renewable Energy Directive (RRED) [12] of the EU, clarify the challenging interplay. As explicitly stated in the WFD, the continuity of sediments is a prerequisite for a "very good ecological status". Sediment continuity is also required as a target to reach a "good ecological status" (depending on the biotic consequences). Sediment connectivity has also come under scrutiny on a global scale to restore ecosystem services [1,4]. Sediment alteration will become a significant water management issue in the next Danube River Basin Management Plan [5]. On the other hand, hydropower is a sustainable energy source with a low carbon footprint and should thus be promoted. In December 2018, the RRED came into force promoting the EU's efforts to meet the emissions reduction commitments under the Paris Agreement. The renewable energy target requires that at least 32% of the EU's total energy needs will be provided by renewables until 2030, including an option for upwards revision by 2023. Finding a balance between these often-conflicting aspects is challenging [3,13,14]. In the past, the operating rules of RoRs were designed to maximize energy revenue neglecting sediment connectivity and continuity. Attempts have been made to change the operating rules of an existing RoR plant in the River Po, Italy, based on an optimization algorithm to maximize hydropower revenue and minimize riverbed degradation [13].

Appropriate sediment management strategies have to focus on (i) maintaining flood safety particularly at the head of the reservoir and (ii) maintaining or re-establishing sediment connectivity. A promising sediment management strategy could be to drawdown reservoirs at natural flood events of high probability to enhance sediment connectivity [3,10,14]. This approach is based on the strongly non-linear relationship between flow and solids transport capacity [15]. In the framework of the EU project "Share", a multi criteria approach was developed to assess flushing strategies on four levels: (i) energy, (ii) economy, (iii) environment and (iv) other criteria such as tourism, landscape, fishing and risk [16]. At some RoRs, considerations to improve sediment connectivity have changed the operating rules from flushing a reservoir as rarely as possible to flushing an RoR reservoir frequently to reduce ecological impacts, such as turbidity for each flushing event [14]. In this respect, flow conditions corresponding to 70% of a 1-year flood ($0.7 \times HQ_1$) are referred to as a good choice for initiating a flushing event [17]. In order to derive appropriate sediment management strategies for RoRs, however, it is necessary to improve the understanding of the transport and sorting processes of non-uniform sediments, in particular, at the head of the reservoir.

Most research and field reports on reservoir sedimentation and delta formation focus on large dams with large storage capacities, e.g., [18–20]. For such storage reservoirs, very fine materials, muddy cohesive deposits and turbidity currents are the dominating processes. For medium-sized gravel bed rivers, bed load is the most significant mode of sediment transport. It is therefore necessary to carefully distinguish between small and large reservoirs and low-head and high-head hydropower, respectively. Therefore, the focus of this paper was to gain knowledge on the delta formation process specifically for low-head RoRs in medium-sized gravel bed rivers.

Although sediment transport and sorting processes are well understood in general terms, a universally valid sediment transport model for nonuniform sediments does not yet exist. In experimental studies, the use of nonuniform sediments is often neglected for simplicity and because scaled sediments become cohesive, even if the corresponding fraction at the prototype scale is non-cohesive.

The objectives of the present paper are (i) to experimentally study sediment transport and sorting processes in the reservoir of a low-head RoR during turbine operation, (ii) to perform a supportive analysis using different bed-load transport models for both uniform sediments and sediment mixtures, (iii) to characterize the delta formation at the head of a reservoir and (iv) to discuss consequences of the results for HPP operators. The delta formation process was studied under the condition that the reservoir was not drawn down, i.e., the operation level of the RoR was kept constant. The process of delta degradation in the case of a reservoir draw down is described and discussed in Reiterer et al. [21].

## 2. Idealized Gravel Bed River and Experimental Setup

### 2.1. Idealized Medium Sized Gravel Bed River

For the experiments, the Muerz river in Styria, Austria, was chosen as a representative medium-sized gravel bed river. The Muerz river is situated north of the Central Alps and belongs to the Danube river system. It is 83 km long and has 23 RoRs and 6 diversion plants along its course, with hydraulic heads ranging from 3 to 10 m [3]. Hydrological and hydro-morphological parameters were chosen based on the Muerz river close to the gauging station Kapfenberg-Diemlach (480 m a.s.l.), which is located about 2.8 km upstream of the river's entry to the Mur river. In Table 1 the hydrological conditions of the Muerz gauging station, Kapfenberg-Diemlach, are summarized. $Q_d$ is the design flow for the RoR Kapfenberg, which is located 2 km downstream of the gauging station. In addition, the catchment area, the river width $W_0$ and the bed slope $S_0$ are provided.

**Table 1.** Hydrological characteristics and grain sizes at the gauging station Kapfenberg-Diemlach.

| River Parameters | Flow Rate (m³/s) |
|---|---|
| Catchment area 1364.5 km² | MQ = 22 |
| River width $W_0$ = 20 m | $Q_d$ = 35 |
| Bed slope $S_0$ = 0.005 | $HQ_1$ = 104 |

MQ = mean flow; $Q_d$ = design flow; $HQ_1$ = 1-year flood.

Figure 2a illustrates the hydrograph of the gauging station Kapfenberg-Diemlach for the years 1971–2015. Average daily flow rates are shown. The horizontal lines indicate MQ, $0.7 \times HQ_1 = 73.5$ m³/s and $HQ_1$, respectively. A flow rate larger than $0.7 \times HQ_1$ occurred on 254 days within 45 years, or 5.6 times a year on average. In Figure 2b, an isolated flow event $> 0.5 \times HQ_1$, with a peak flow of approximately $0.7 \times HQ_1$, is shown, which has a duration of 7.5 days. This event is approximated by a step function of five steps, each one lasting for 36 h. The steps represent the following fractions of $HQ_1$: 0.5–0.6–0.7–0.6–0.5. These flow rates were used in the experiments.

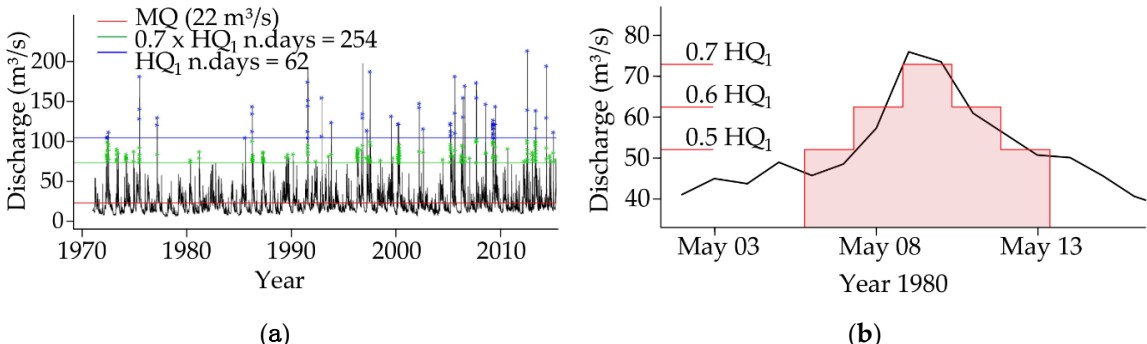

(a)                    (b)

**Figure 2.** Hydrograph of the gauging station Kapfenberg-Diemlach in the Muerz River in Styria, Austria: (**a**) average daily flow rate for the years 1971–2015; (**b**) isolated flow event at the station Kapfenberg-Diemlach in May 1980, with a duration of 7.5 days, approximated flow event in steps of 0.5–0.6–0.7–0.6–0.5 × $HQ_1$. Hydrograph data retrieved from www.ehyd.gv.at on 10 March 2019.

### 2.2. Experimental Setup

The experiments were conducted at the Hydraulic Engineering Laboratory of the University of Natural Resources and Life Science Vienna, Austria. The physical model was built at a scale of 1:20 in a 1.33 m wide and 14 m long straight flume consisting of an inlet basin with vertical and horizontal flow straighteners, a sediment feeder, a 9.5 m long experimental section, a sand trap with a steel box to collect and weigh the entrained sediments and a bottom hinged tail gate to manipulate the water level (Figure 3). The water came from the central water circulation system of the lab and was adjusted by frequency-controlled pumps. A magnetic flow meter was used to measure the discharge. Five

ultrasonic probes were available to record the water levels during the experiments. The probes were mounted on a linear positioning system (Figure 3b). It was possible to measure longitudinal water surface profiles and to record the water levels over time at single points.

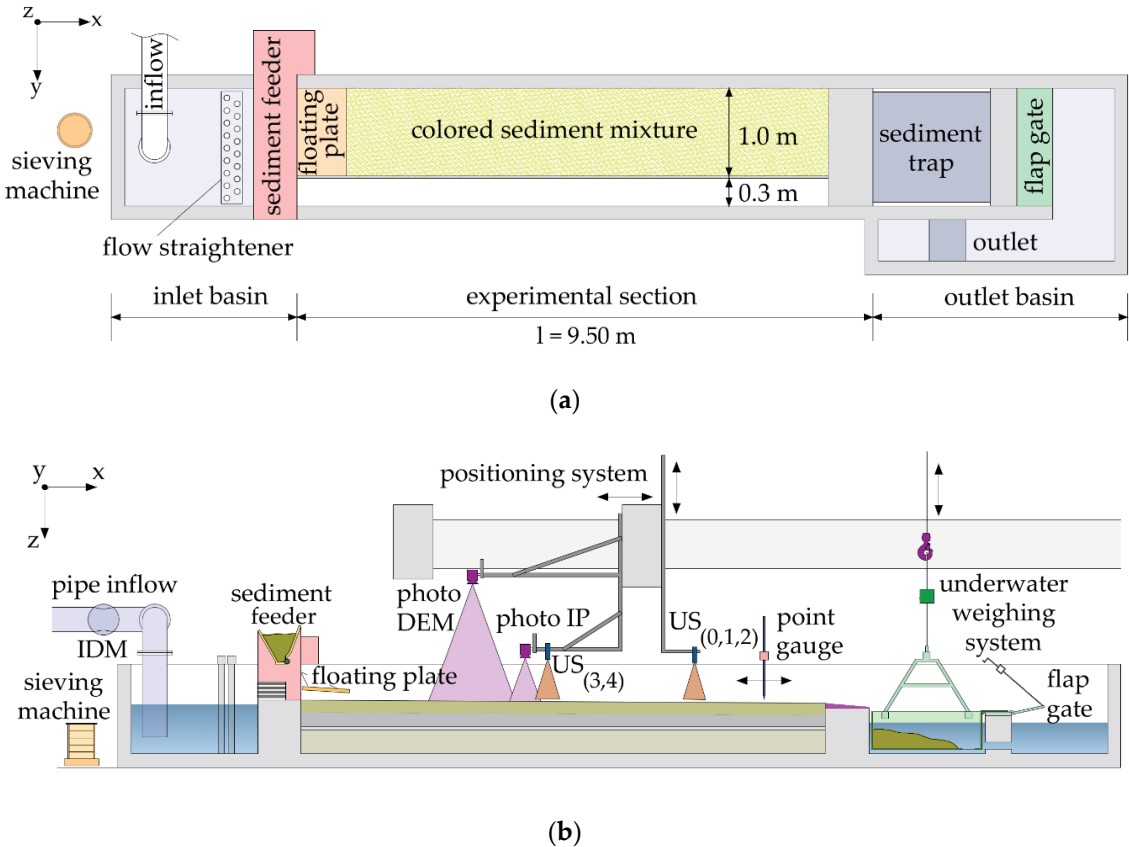

(**a**)

(**b**)

**Figure 3.** Experimental setup, (**a**) plan view; (**b**) longitudinal section of the straight flume. DEM = digital elevation model, IP = image processing, US = ultrasonic probe.

Sediment was added to the model via a sediment feeder. The feeder consisted of a sediment hopper with a bottom slit spanning the entire width. A rotating horizontal shaft was situated below the bottom slit of the hopper. The shaft had eight longitudinal slots to gather sediments from the hopper while rotating. The shaft was driven by a speed-controlled motor. With an accuracy of 5%, a calibration could be established relating the driving speed (Hz) to the sediment feeding rate (kg/h). Bed levels were recorded by means of photogrammetry. Ground control points (GCP) on either side of the physical model were surveyed at an accuracy of 0.1 mm. Pictures of the sediment distribution were taken when the model was drained from water. A Nikon D7100 camera (lens: AF NIKKOR 20 mm, F/2.8D, resolution: 6000 × 4000 pixels) was mounted on the positioning system. The camera reached several positions along the channel axis where pictures were taken. Two subsequent pictures had an overlap of 80%. On each picture at least one GCPs was visible. The software Agisoft Metashape 1.5.2 automatically detected the GCPs on the pictures via a unique circle code. From the pictures, an orthophoto and a digital elevation model (DEM) of the bed levels was computed with a resolution of 12 mm and an accuracy in vertical direction of +/−1 mm. Further, by aligning the pictures for the DEM with 60 additional high-resolution images, an orthophoto with a pixel size of 0.1 by 0.1 mm was created.

*2.3. Sediment Mixture and Analysis*

2.3.1. Sediment Scaling

When sediment transport processes are investigated in a scaled model, it is crucial to ensure that incipient sediment motion and sediment transport rates are similar in nature and in the scaled model. This similarity is obtained when the Shields parameter $\theta$ and the particle Reynolds number Re* are the same in the model and in nature [22]. The Shields parameter is the dimensionless shear stress calculated as $\theta = \tau/((\varrho_S - \varrho)\, g\, d_{50})$, where $\tau = \varrho\, g\, h\, S_E$ = shear stress (N/m²), $\varrho_S$ = sediment density (kg/m³), $\varrho$ = water density (kg/m³), h = water depth (m) and $S_E$ = energy slope (-). In our experimental study, the relative sediment density $\varrho_r = (\varrho_S - \varrho)_N/(\varrho_S - \varrho)_M$ is assumed to be the same in nature and model. This assumption is justified because the density of the model sediments lies in the range of the sediment densities of the Muerz river. For an undistorted model with length scale $l_r$, a similarity of the Shields parameter implies that the particle diameter scale equals $l_r$. According to [22] the similarity of Re* can be neglected, if Re* > 100. Based on the mean diameter $d_m$ of the model sediments, Re* > 150, even for the lowest tested flow rate of 0.5 × HQ₁. Under these assumptions, the time scales $t_r$ of the flow and of the moving sediments are identical, i.e., $t_r = l_r^{0.5}$.

2.3.2. Sediment Mixture

The model sediment consisted of five different grain size classes of dyed quartz sand, having a density of $\varrho_s = 2600$ kg/m³ and a bulk density of 1700 kg/m³. The grain size classes of the 1:20 scale experiments and the corresponding grain sizes at 1:1, as well as the initial mass fraction and the color of each class are summarized in Table 2.

**Table 2.** Grain size classes of the 1:20 scale experiments and corresponding grain size classes at 1:1, initial mass fraction (%) and non-uniformity parameters [1] U and $\sigma$.

| Fraction No. | Grain Size 1:20 (mm) | Grain Size 1:1 (mm) [2] | Color | Initial Mass Fraction (%) |
|:---:|:---:|:---:|:---:|:---:|
| 1 | 0.7–1.2 | 14–24 | Yellow | 15 |
| 2 | 1.2–2.0 | 24–40 | Red | 15 |
| 3 | 2.0–3.0 | 40–60 | Green | 20 |
| 4 | 3.0–4.0 | 60–80 | Black | 25 |
| 5 | 4.0–6.0 | 80–120 | White | 25 |

[1] The nonuniformity parameters are U = $d_{60}/d_{10}$ = 3.29 mm and $\sigma = (d_{84}/d_{16})^{0.5}$ = 2.25 mm. [2] $d_{30}$ = 40 mm, $d_m$ = 60 mm, $d_{90}$ = 102 mm.

2.3.3. Grain Size Distributions from Mass Fractions and from a Novel Surface-Based Method

The experiments were conducted under mobile-bed conditions, i.e., the bed material was mobile and sediments were fed to the model. At the downstream end of the experimental section, a steel box in the sediment trap was used to collect the entrained sediments. After each test run, which lasted for one or two hours, the sediment in the steel box was weighed under water by means of a crane scale (MCWN-Ninja from DINI ARGEO, accuracy +/−0.5 kg). The sediment in the steel box was subsequently dried and sieved resulting in the grain size distribution of the transported material based on mass fractions.

To determine the grain size distribution of the bed surface material, an image processing method was developed making use of the fact that each grain size class had a different color. In order to ensure that each grain of the sediment mixture was represented by several pixels, high resolution images (>300 ppi = pixels per inch) were required. A 30 W LED working lamp, mounted on the positioning system, was used to illuminate the bed surface as evenly as possible. A photogrammetric survey was carried out using the same GCPs as described in Section 2.2. The pictures of the bed surface were orthorectified and merged to obtain a single orthophoto of the bed surface (Figure 4a). Each pixel of the compound orthophoto was subsequently assigned to one of the five color classes.

Due to different light intensities and shadow formation, the color of each sediment fraction was represented by different shades of the respective color. An image processing method was required which reduced the different color shades to five different colors only. Therefore, for each pixel, a maximum likelihood classification, including predetermined signature classes, was applied, which was available in the software ArcMap 10.5. Figure 4a illustrates the compound orthophoto of the experimental section covered with the colored sediments ($9.2 \times 1$ m$^2$). Contour lines are illustrated in black. The orthophoto represents the water-worked bed after a flow rate of $0.7 \times HQ_1$. The flow direction is from left to right. Different areas are discernible with the naked eye, where different colors are dominant. At this flow rate sediments are transported as dunes. Three of these dunes are marked with blue lines. The red square indicates an arbitrary section of the bed surface, where the image processing method is further illustrated. Figure 4b represents the zoomed red square of the orthophoto, Figure 4c its reduction to five different colors only and Figure 4d the resulting grain size distribution.

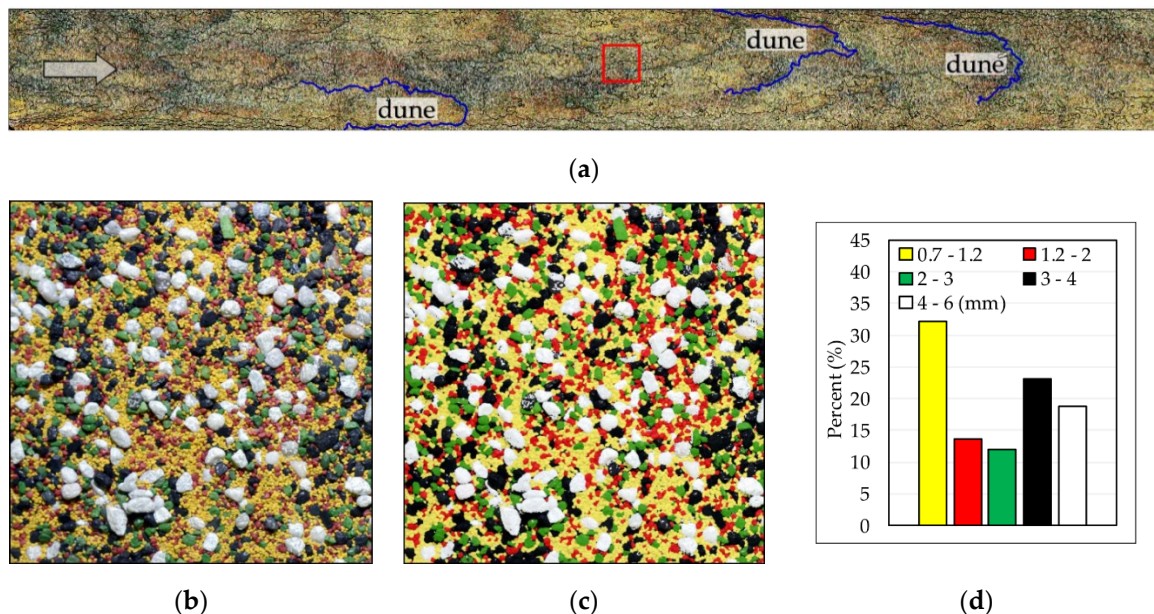

**Figure 4.** (**a**) Compound orthophoto of the bed surface material of the experimental section. The arrow indicates the flow direction. The red square indicates the location of the $0.12 \times 0.12$ m section detail; (**b**) red square detail of the orthophoto; (**c**) red square detail of the processed orthophoto which is reduced to five different colors only, each pixel belongs to one and only one color class; (**d**) grain size distribution of the red square detail.

The developed image processing method allowed us to determine the grain size distribution (GSD) of the bed surface material. Unlike the grain size determination of a sieve analysis, which yields mass fractions, the grain size distribution from image processing provides an area-measured distribution. The question arises if or how these two methods differ from each other. To answer this question, separate tests were carried out where the surface material of a test area was analyzed with both methods. It turned out that the area-based method underestimates the coarse fractions compared to the mass-based method. The reason for this is that the mass-based method considers a grain in three dimensions, while the area-based methods treats the grains as 2D-objects. A comprehensive description of the image processing method and its error estimation will be described in a follow-up paper (in prep.). In this paper, GSDs are only compared with each other if they were obtained from the same method.

### 2.3.4. Sediment Transport Analysis

Prior to the experiments, the sediment transport capacities were determined from five models and were later compared with the measured transport rates. Specifically, the following models were

used: (i + ii) the formula of Meyer-Peter and Müller (MPM) [23] with two different constant factors, labelled "MPM-5" and "MPM-8"; (iii) the formula of Smart and Jäggi [24]; (iv) the model of Wu et al. [25] for nonuniform sediment transport in alluvial rivers, "WU"; and (v) the surface-based transport model for sediment mixtures developed by Wilcock and Crowe [26], "WC".

Like most other sediment transport formulas, the MPM formula can be expressed in dimensionless form, where the dimensionless sediment transport rate is a function of the difference of the actual bed shear stress and the critical bed shear stress. In this dimensionless notation, the formula includes a constant factor 8: see, e.g., [24]. Some researchers suggest that a factor of five is more appropriate to match experimental or field data, e.g., [27]. The labels MPM-8 and MPM-5 thus indicate which factor was used.

The MPM formula uses the mean grain diameter $d_m$ as a characterization of the sediment mixture. In addition to the mean diameter $d_m$, the SJ formula considers the non-uniformity of the sediments by introducing a factor $(d_{90}/d_{30})^{0.2}$, where $d_{90}$ and $d_{30}$ are the grain diameters for which 90% and 30% of the sediment mixture are finer. Both approaches calculate a single transport rate for the entire sediment mixture. In contrast, the models of WU and WC provide transport rates for each grain size class. The model of WU considers the hiding and exposure mechanism of nonuniform sediment transport. The probabilities for a grain size class to be exposed or hidden is supposed to be stochastically related to the size and gradation of sediment mixture. From these probabilities a correction factor is developed [25]. Similar to most other mixed-sediment transport models, e.g., [25,28,29], the transport model of Wilcock and Crowe incorporates a hiding-exposure function which decreases the shear stress for finer fractions and increases the shear stress for coarser fractions. According to their experimental observations, the hiding function considers the non-linear behavior of sand content on the gravel transport rate. Specifically, the sediment transport rate follows approximately a bi-modal log-linear function [26].

*2.4. Experimental Procedure*

In Figure 1 a longitudinal sketch of an RoR is illustrated where three sections are plotted: (i) a free-flowing section Sec 1, which is not affected by the backwater of the dam, (ii) the head of the reservoir (Sec 2), where the backwater effect starts, and (iii) a section Sec 3 immediately upstream of the dam structure. Assuming a representative dam height of $h_d = 5$ m for a typical RoR located at the Muerz River, the reservoir length $l_{res}$ can be calculated from the bed slope $S_0$, the water depth $h_0$ of the free-flowing section and the water depth and $h_d$ at the dam [3].

The experiments for studying delta formation were divided into two major test series. First, the sediment transport capacity for a free-flowing section was examined for different flow rates (Sec 1 in Figure 1). Second, the sedimentation process during turbine operation was determined at the head of a reservoir where the backwater starts (Sec 2 in Figure 1). The backwater curve approaches the free-flowing upstream river section asymptotically. A meaningful criterion to determine the location where the backwater effect is negligible is where the water depth h does not exceed the water depth of the free-flowing section $h_0$ by more than +1% [30,31]. In the present study, the head of the reservoir was thus defined as the location where $h = 1.01 \times h_0$. In the physical model, the backwater effect was generated by the bottom-hinged flap gate at the downstream end of the model. The longitudinal station of the reservoir head could thus be chosen independently. In order to verify if the water-surface levels generated by the flap gate resemble a real situation the backwater curve was calculated step by step in upstream direction by using the tables of Rühlmann: see, e.g., [32]. The present paper deals with Sections Sec 1 and Sec 2. It forms part of a larger study on RoR reservoir sedimentation and desiltation using the same bathymetric information and technical details since 2017 and will be extended for future research on RoR as well.

**3. Results**

First, the results of the free-flowing section, Sec 1, are presented in Section 3.1. The conditions for dynamic equilibrium, sediment transport rates and grain size distributions for the different flow rates are described. Measured vs. calculated results are compared. The determined sediment

transport rates and grain size distributions are further used as independent parameters for the experiments at the head of the reservoir (Sec 2), which are presented in Section 3.2. If not otherwise stated, all results in this section are presented in model dimensions (1:20).

*3.1. Characteristics of the Free Flowing Section*

3.1.1. Sediment Transport Rates for the Free-Flowing Section

For each flow rate, $0.5 \times HQ_1$, $0.6 \times HQ_1$, $0.7 \times HQ_1$ and $0.8 \times HQ_1$, the conditions for dynamic equilibrium, were determined iteratively by keeping the bed slope of 0.005 constant and by varying the sediment feeding rate. In addition, the GSD of the added sediment was adopted step by step to align the GSDs of the sediment input and sediment output. For all flow rates, a dynamic equilibrium was found where bed and water levels were virtually parallel along the channel (Table 3). Except for the third experiment of $0.7 \times HQ_1$, water surface slopes were measured with a point gauge at 11 cross sections along the channel. For the third experiment of $0.7 \times HQ_1$, two continuous longitudinal water surface profiles were measured by means of two ultrasonic probes.

**Table 3.** Transport rates, ratio of fed sediment (input) vs. transported sediment (output), bed and water surface slopes at dynamic equilibrium for the investigated flow rates.

| Flow Rate | Duration | Rep. nbr. | Transport Rate (kg/hm) | Input / Output (-) | Bed Slope (-) | Water Surface Slope (-) |
|---|---|---|---|---|---|---|
| $0.5 \times HQ_1$ | 8 h | 1 | 7 | 0.93 | 0.0046 | 0.0047 |
| $0.6 \times HQ_1$ | 8 h | 1 | 27 | 1.04 | 0.0045 | 0.0050 |
| $0.7 \times HQ_1$ | 8 h | 1 | 51 | 0.98 | 0.0048 | 0.0048 |
| $0.7 \times HQ_1$ | 8 h | 2 | 52 | 1.02 | 0.0048 | 0.0049 |
| $0.7 \times HQ_1$ | 8 h | 3 | 53 | 0.98 | 0.0049 | 0.0048 |
| $0.8 \times HQ_1$ | 4 h | 1 | 117 | 1.09 | $0.005 \pm 10\%$ | $0.005 \pm 10\%$ |

Prior to the experiments, sediment transport rates, as well as grain size distributions of mobile sediments, were calculated for the free-flowing section, Sec 1. The sediment mixture for the calculations is given in Table 2. Without knowing the water depth in advance, the factor of grain roughness vs. channel bed roughness (used in the models of MPM, SJ and WU) was not considered. The angle of repose in air and underwater, as used in the model of SJ, was estimated, rather than determined experimentally. The idea was to assess the predictability of sediment transport rates without having information of the angles of repose and of the ratio of grain vs. channel bed roughness. In Figure 5, the calculated sediment transport rates for discharges ranging from $0.5 \times HQ_1$ to $0.9 \times HQ_1$ are illustrated. The predicted transport rates are similar for WC and SJ. The rates of WU are higher but have about the same gradient. The model of MPM-5 has a much steeper gradient. Per definition, the gradient of MPM-8 is 1.6 times larger than that of MPM-5. In Figure 5, the range between $0.5 \times HQ_1$ to $0.7 \times HQ_1$ is shaded in grey because this range is often recommended as threshold criterion for starting a reservoir drawdown [15].

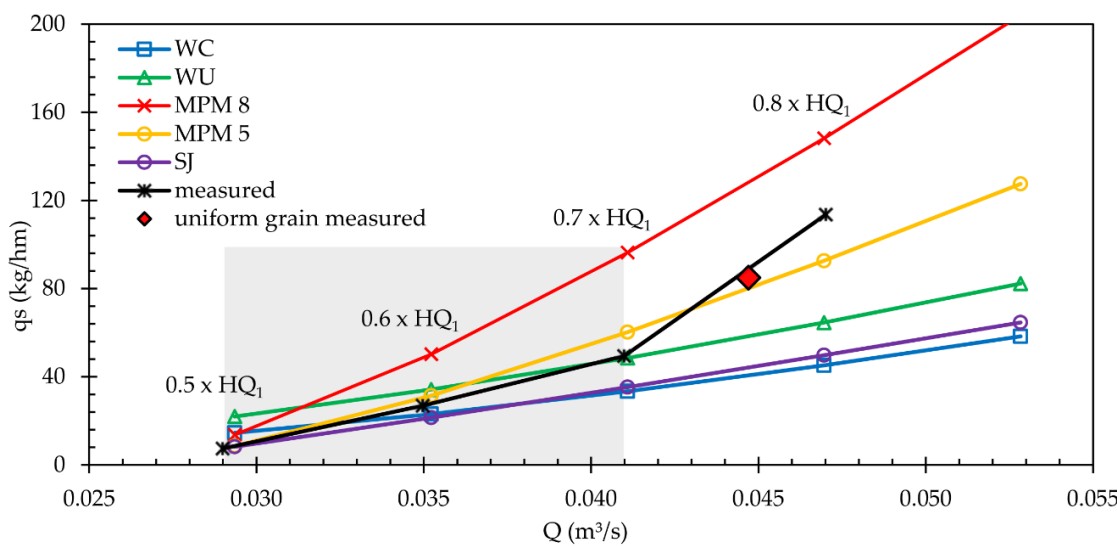

**Figure 5.** Sediment transport rates flow rates ranging from $0.5 \times HQ_1$–$0.9 \times HQ_1$: calculated sediment transport rates according to models of MPM, SJ, WC and WU. Measured transport rates of present study (black line) and of a previous study under similar conditions (red diamond) [3].

### 3.1.2. Measured vs. Calculated Transport Rates

The measured transport rates show a bimodal trend (Figure 5). They are approximately linear for discharges ranging from $0.5 \times HQ$ to $0.7 \times HQ_1$. Then, there is a sudden increase in transport rates for $0.7 \times HQ1$. The measured transport rate for uniform sediment ($d_{50}$ = 2.6 mm, $\sigma$ = 1.17, S = 0.005) from a previous study [3] fits well to the steep side of the transport curve for discharges $\geq 0.7 \times HQ_1$. Neither of the sediment transport models predicts the bimodal trend of the experiments. The transport rates of MPM-5 have the best least squares fit. In Table 4, the calculated vs. measured total sediment load for the approximated flood event of Figure 2b is summarized. The results are given in a dimension of 1:1. The total load was calculated for flow rates of $0.6 \times HQ_1$ and $0.7 \times HQ_1$ lasting for 72 and 36 h, respectively. Transport rates for $0.5 \times HQ_1$ were not considered in the calculations assuming that in this period the discharge is not free-flowing because of gate operations restricting the flow. Thus, the transport rates can be neglected. For the considered flood event, the transport rates of WU deviate by 13% from the measured transport rate. Except for MPM-8, all calculated sediment loads are within +/− 25% of the measured transport rate.

**Table 4.** Calculated and measured total sediment load (kg × 10³) for the step-wise flood event of Figure 2b in a dimension of 1:1.

| SJ | WC | WU | MPM-5 | MPM-8 | Measured |
|---|---|---|---|---|---|
| 252 (−24%) | 256 (−23%) | 376 (+13%) | 396 (+19%) | 517 (+56%) | 332 |

For flow rates $< 0.7 \times HQ_1$, the gradient of the transport rates is small. For flow rates $\geq 0.7 \times HQ_1$, the gradient of the transport rates is almost three times larger. Therefore, for this particular case, $0.7 \times HQ_1$ is the turning point where reservoir drawdown might be considered. This is why this flow rate was chosen to study delta formation to assess pros and cons of a drawdown.

### 3.2. Delta Formation at the Head of the Reservoir

The delta formation tests were carried out three times under the condition that the reservoir was not drawn down. They will be referred to as D1, D2 and D3, respectively. The experimental conditions were the same for all test runs, except for the location of the head of the reservoir. By means of the flap gate at the downstream end of the physical model, the location of the reservoir head was adjusted. For D1, D2 and D3, the location of the reservoir head was at X = 2.5 m, X = 1.5 m and X

= 4.0 m, respectively, where X = 0 m is at the beginning of the experimental section which is 9.5 m long. Therefore, for D1 and D2, the reservoir head is located in the upper quarter of the experimental section and for D3 the reservoir head is about halfway of the experimental section's length. For D1, water levels were measured with a point gauge. D2 and D3 were carried out by measuring the water levels with two ultrasonic probes which were aligned in lateral direction at Y = 0.33 m and Y = 0.66 m, respectively. The probes were mounted on the positioning system and recorded longitudinal water surface profiles. All delta formation tests were carried out at a flow rate of $0.7 \times HQ_1$ because this flow rate was determined to be the turning point of the bimodal transport curve (Figure 5).

### 3.2.1. Sediment Transport Rates at the Head of the Reservoir

In Table 5, sediment feeding rates (input) and measured transport rates (output) are summarized for all delta formation test runs. For D1 and D2, almost all fed sediments accumulated in the experimental section and were not transported to the end of the physical model. A very small fraction of 1%–3% actually reached the sand trap. About 80% of the material in the sand trap belonged to the two smallest grain size classes. For test run D3, similar transport rates were measured for the first four hours. The transport rate increased considerably for the final four hours of the experimental time. The reason for this is the location of the reservoir head which was located in the middle of the experimental section for D3, while the reservoir head was located further upstream for D1 and D2. As will be explained in more detail in the following section, the delta front moved out of the physical model after two hours of experimental time. Thus, for the remainder of the D3 experiment, the physical model represented the tail part of the delta formation, which had steeper energy gradients and thus higher transport rates. In addition, for the material in the sand trap, the fractions of the coarser grain size classes increased compared to D1 and D2. This means that, due to the sedimentation, the impounded section close to the reservoir head has smaller water depths and higher shear stresses which are able to transport coarser fractions than for the test cases D1 and D2.

**Table 5.** Delta formation test runs D1–D3: transport rates, ratio of fed sediment (input) vs. transported sediment (output), transported fractions of the yellow (0.7–1.2 mm) and red (1.2–2 mm) grain size class after 8 h.

| Test Run | Input (kg/h) | Output 0-4 h (kg/h) | Output 4–6 h (kg/h) | Output 6–8 h (kg/h) | Yellow Fraction (%) | Red Fraction (%) |
|---|---|---|---|---|---|---|
| D1 | 45 | 1.5 | 1.5 | 1.3 | 46 | 36 |
| D2 | 52 | 0.5 | 0.35 | 0.20 | 45 | 35 |
| D3 | 52 | 1.4 | 11.5 | 35.5 | 15 | 23 |

### 3.2.2. Water and Bed Surface Evolution at the Head of the Reservoir

In Figure 6, longitudinal water and bed surface profiles are illustrated at the head of the reservoir for test run D2. The flap gate at the downstream end of the physical model was adjusted such that the reservoir head was located at X = 1.5 m. The bed surface at the beginning of the test run represented the bed levels of the free-flowing section (Section 1) in a state of dynamic equilibrium where bed and water surface levels are parallel. The water levels of the Section 1 are shown in grey for comparison. The black line illustrates the measured water surface levels of the backwater curve at the beginning of the experiment (0 h). The purple dashed-dot line represents the calculated backwater curve (BWC), showing good agreement with the measured BWC. The measured backwater curve is virtually horizontal from X = 4.0 to X = 8.5 and approaches the water levels of the free-flowing section further upstream. Due to the backwater effect the transport is decelerated. Sediments accumulate in upstream and downstream direction from the reservoir head. The delta formation is clearly visible from these measurements. Different areas characterizing the delta formation are striking: a steep front termed "delta front", followed by an almost horizontal area termed "delta top" and a "delta tail". During the first two hours, the front of the delta formation

travels until X = 6.5 m. Over time, the delta front travels shorter distances in equal time intervals (2 h) but increases in height. After eight hours of experimental time, the delta front reaches the end of the experimental section at X = 8.5 m. It has a height of 0.031 m and the slope is about 35%. The delta top is almost horizontal.

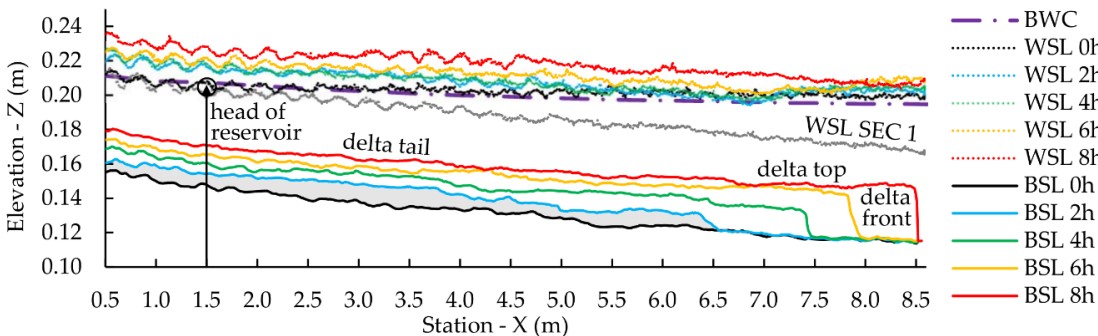

**Figure 6.** Bed and water surface profiles for test run D2 at the head of the reservoir at a flow rate of $0.7 \times HQ_1$ during turbine operation. The grey line represents the water surface profile of the free-flowing section which corresponds to the bed levels at hour 0 (black line). The head of the reservoir is located at X = 1.5 m. Different areas of the delta formation, namely "delta front", "delta top" and "delta tail", are indicated. WSL = water surface levels, BSL = bed surface levels, BWC = calculated backwater curve.

In Figure 6, test run D2 is shown because detailed water surface profiles were measured for this case. For test run D1, only point gauge measurements were available. The results of D1 and D2 are comparable regarding delta formation and water surface evolution.

For the third test case, D3, the reservoir head was located at X = 4.0 m. The focus of this test was to study the delta formation process in upstream direction. The delta front reached the downstream end of the experimental section after 2 h. Thus, the delta front could not be observed any longer for the following 6 h. The bed and surface levels upstream of the delta front revealed another interesting process. The bed levels and water surface levels increased evenly in longitudinal direction, building bed and surface profiles almost parallel to that of the free-flowing section only at a higher elevation (Figure 7).

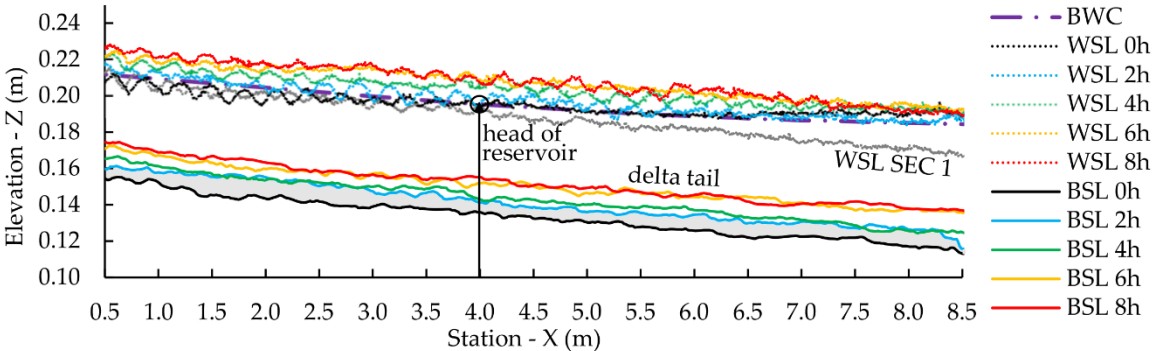

**Figure 7.** Bed and water surface profiles for D3 at the head of the reservoir at a flow rate of $0.7 \times HQ_1$ during turbine operation. The profiles were measured every two hours. The grey line represents the water surface profile of the free-flowing section which corresponds to the bed levels at hour 0 (black line). The head of the reservoir is located at X = 4.0. WSL = water surface levels, BSL = bed surface levels, BWC = calculated backwater curve.

Figures 6 and 7 suggest that bed levels of the delta tail become parallel to the initial bed levels. Based on Table 6, this observation can be further refined. Table 6 summarizes bed, water and energy slopes for delta tail, top and the section downstream of the delta, respectively. The slopes of test runs

D2 and D3 are listed after four and eight hours of experimental time. The slopes for delta top, front and the section downstream of the delta are given only for D2 because for D3 the delta top had already moved past the experimental section. In addition, the dimensionless Shields parameter θ is given. It is used to determine the conditions for sediment transport and to compare it with the experimental conditions. For the calculation of the Shields parameter, an error propagation analysis was carried out. Realistic uncertainties/errors of the measured quantities were defined. For six test runs of Table 6 the propagated error was less than 10%. The only exception was the D2 delta top test run after 4 h. The energy slope was almost zero for this case. Therefore, the estimated error was huge. The calculated Shields parameter θ = 0.0003 agrees well with the experimental observations that there was no entrainment on the delta. Nonetheless, this calculated value must be treated with caution because of the error analysis.

**Table 6.** Delta tail and top after four and eight hours: bed, water and energy slopes in percent (%), negative signs indicate rising slopes, Shields parameter θ.

| Test Run | Hour | Delta Section | Bed Slope | Water Slope | Energy Slope | θ |
|----------|------|---------------|-----------|-------------|--------------|-------|
| D2 | 4 | tail | 0.56 | 0.38 | 0.53 | 0.065 |
| D2 | 4 | top | 0.30 | −0.38 | 0.00002 | 0.0003 |
| D2 | 4 | downstream | 0.32 | 0.08 | 0.14 | 0.024 |
| D2 | 8 | tail | 0.52 | 0.37 | 0.49 | 0.061 |
| D2 | 8 | top | 0.06 | 0.29 | 0.13 | 0.017 |
| D3 | 4 | tail | 0.45 | 0.37 | 0.45 | 0.051 |
| D3 | 8 | tail | 0.47 | 0.47 | 0.47 | 0.051 |

After eight hours of experimental time, bed and energy slopes of the delta tail are close to the dynamic equilibrium slope $S_0$ of the free-flowing section for D2 and D3. For D2, the water slopes are flatter than $S_0$, indicating gradually decelerating flow on the delta tail due to the backwater of the delta top. As expected, the energy slopes are smallest on the delta top. After four hours, the water surface rises and the energy slope is close to zero. At this particular instant of time, the shear stress on the delta is two orders of magnitude below the critical shear stress. After eight hours, the bed surface is almost horizontal, but due to the increased sedimentation on the delta top, the energy slopes are higher than after four hours. This reveals a continuous process on the delta top. As sediments accumulate on the delta top the energy slope increases. At some instance of time, the bed shear stress which is related to the energy gradient exceeds the critical shear stress such that sediments on the delta are re-mobilized. During this period, the delta moves further into the reservoir.

By combining D2 and D3, the following summary can be drawn for a 0.7 × HQ₁ flood event which lasted for 36 h. All dimensions are given at full scale (1:1). The delta formation which developed moved 120 m into the reservoir. After the flood event, the delta formation was responsible for a water level rise which reached as far as 100 m into the reservoir. At 70 m upstream of the reservoir head, the bed levels were parallel to the initial bed levels of the free-flowing section only at a higher elevation (+0.4 m). Similarly, the water levels rose by 0.4 m. Presumably the delta tail extension is longer than 70 m. However, the experimental section was not long enough to capture the entire delta tail and to locate the upstream position where the delta formation merged into the initial river bed. This terminal section will be termed the "delta appendix". Given that the delta tail is virtually parallel to the initial bed, it becomes obvious that for the appendix to merge into the initial river bed the appendix has to be flatter than the initial bed slope. The flatter appendix is visible for test run D2 (Figure 6, blue line) and test run D3 (Figure 7, blue line), where the bed levels after two hours almost reconnect with the initial bed levels of hour 0. Similar observations were made for D1 (not shown).

### 3.2.3. Grain Size Distributions of the Delta Formation

During the delta formation experiments, sediment sorting effects were clearly visible due to the colored sediments. In Figure 8, the grain size distributions (GSD) and the corresponding mean diameters $d_m$ of the delta formation are illustrated for the tail, the top and the front of the delta of test run D2 after four hours of experimental time. As a reference, the GSDs of the total area at the beginning of the experiment (hour 0) and the area downstream of the delta formation are shown. A plan view scheme of the experimental section identifies the different areas.

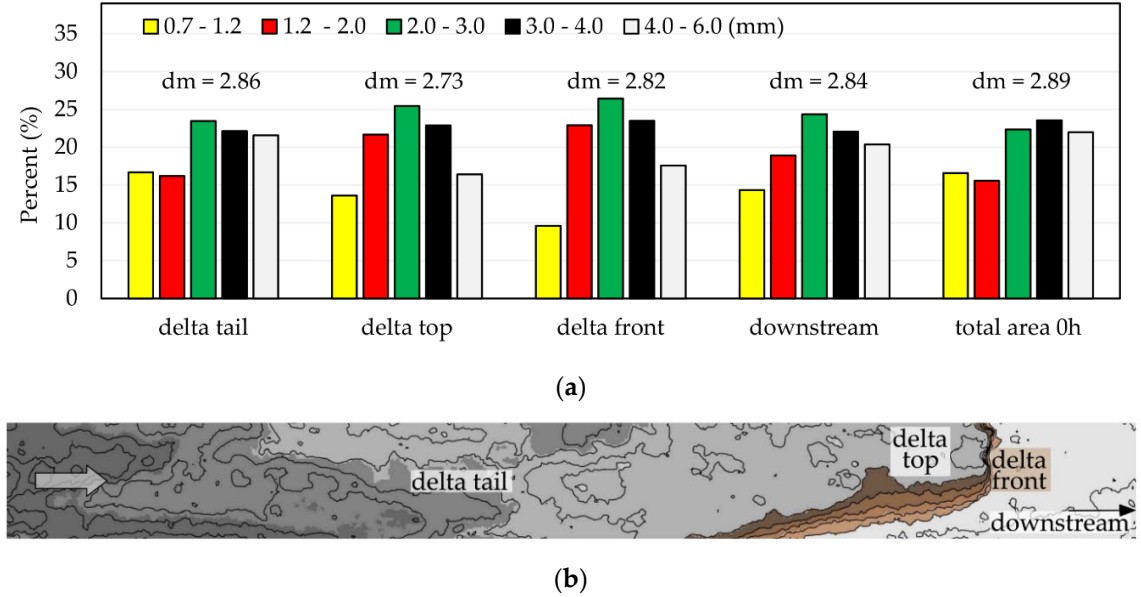

(**a**)

(**b**)

**Figure 8.** (**a**) Grain size distributions of the delta formation for test run D2 after four hours; (**b**) plan view of delta formation indicating the areas of delta tail, delta top, delta front and downstream section, respectively.

Following the above-mentioned finding that the delta tail approaches the dynamic equilibrium conditions of the free-flowing section, the grain size distributions should also be similar. In fact, the delta tail has a GSD which is very similar to that of the total area regarding the distribution and the mean diameter $d_m$. On the delta top, the middle grain size classes red, green and black are dominant. The reduced fraction of the yellow grain size indicates that the shear stresses on the delta top are large enough to transport the yellow grains. This is consistent with the observation that mainly the smallest yellow fraction and parts of the red fraction were transported further downstream (cf Table 3). It also agrees with the measured transported material in the sand trap. The increase of the red fraction on the delta top and the measured transported red grains in the sand trap suggest that parts of the red material settled on the delta top, while other parts were transported further downstream. In total, the mean diameter $d_m$ on the delta top decreased. This might be counter-intuitive at first glance. It might appear that the delta top should be coarser because the larger fractions settle due to the reduced bed shear stress (cf Table 6). A plausible explanation might be that the largest white fraction was overlapped by the red, green and black fractions. In other words, the observed finer GSD on the delta top could be explained by vertical sorting effects.

The delta front contains a small fraction of yellow grains. Experimental observations suggest that the grains of this grain size class are kept in suspension as they pass the delta front because of locally increased velocities and turbulent fluctuations. The Rouse number is 1.6 for the yellow fraction indicating suspended load. For all other fractions, the Rouse number is >2.5.

The area downstream of the delta should resemble that of the reference area because there should be no transport due to the back water. The results reveal that the GSD is similar except of the noticeable increase of the red fraction. This might be explained by red fractions, which are

transported over the delta and settle downstream because of further reduced shear stresses. At the same time, the red fractions cover the larger fractions.

### 3.2.4. Measured vs. Calculated Transport Rates at the Head of the Reservoir

Finally, the experimental results are compared with the sediment transport models. As transport rates were only measured at the downstream end of the model, three different areas are compared: (i) the section downstream of the delta after four hours of test run D2, (ii) the delta top of test run D2 after eight hours and (iii) the delta tail of test run D3 after eight hours. For the calculation, the measured water depth and the energy gradient of the respective section were chosen as independent parameters. In Table 7, the results are summarized. After four hours of test run D2, the models of MPM-5 and SJ predict that there is no sediment transport for the section downstream of the delta. The models of WC and WU predict very little sediment transport. Their results are quite close to the measured transport. According to WU, only the two smallest fractions are in motion. The model of WC predicts that the two smallest fractions comprise 65% of the transported material. This is quite close to the experimentally determined 78% of the material captured in the sand trap.

**Table 7.** Calculated and measured transport rates in (kg/hm) of sections which are located at the downstream end of the experimental section.

| Test Run | Hour | Delta Section | θ | MPM-5 | SJ | WC | WU | Exp |
|:---:|:---:|:---:|:---:|:---:|:---:|:---:|:---:|:---:|
| D2 | 4 | downstream | 0.024 | 0 | 0 | 0.12 | 0.14 | 0.4 |
| D2 | 8 | top | 0.017 | 0 | 0 | $5 \times 10^{-3}$ | $4 \times 10^{-6}$ | 0.2 |
| D3 | 8 | tail | 0.051 | 82 | 10 | 14 | 14 | 36 |

For the delta top section of D2 after eight hours, the models of MPM-5 and SJ predict no sediment transport. For the models of WC and WU, the transport rate is nonzero but almost negligibly small. The model of WU predicts that only the smallest fraction is in motion. According to WC, all fractions are in motion which coincides with the experimental observation. WC predicts that 65% of the total sediment load belong to the two smallest fractions. The measured proportion of the two smallest grain size classes was 68%. For the decelerated flow cases, the model of WC is better suited to predict the GSDs of the transported material than the model of WU.

After eight hours of test run D3, the delta tail resembles the free-flowing section. Both models WC and WU predict the GSD of the transported material with reasonable agreement. However, they underestimate the measured transport rate by far. The model of MPM-5 overestimates the transport rate. Summarizing, each of the tested models has its strong and weak points but none of the models is capable to predict all experimentally observed transport phenomena with sufficient agreement.

### 4. Discussion

From a technical point of view, it is crucial to know at what flow rates there are considerable sediment loads to justify a drawdown that is not strictly necessary from a flood prevention perspective. There is not a general answer to this because the process depends on many different factors, such as (i) the sediment availability from upstream, (ii) the geometry of the reservoir, (iii) whether or not the RoR is part of a chain of RoRs and many more.

A focus of this study was to quantify sediment loads during flood flows of high probability and to compare it to five well known bed-load formulas. The formulas of Meyer-Peter and Müller (MPM-5 and MPM-8), Smart and Jäggi (SJ), Wu et al. (WU) and Wilcock and Crowe (WC) were tested for discharges ranging from $0.5 \times HQ_1$–$0.9 \times HQ_1$. None of the models were able to predict the measured bimodal trend of the measured transport rates for the free-flowing section. For the simplified flood event of high probability (Figure 2b), the model of WU predicted the total sediment load best. Considering the whole measured discharge range, the formula MPM-5 has the best least squares fit. Except for MPM-8, all calculated transport rates showed reasonable to good agreement under uniform flow conditions (Table 4)

The comparison between calculated and measured transport rates and GSDs on the delta (Table 7) once again proves the deficiencies of sediment transport models. Most of them were derived under the assumption of good agreement between the particle composition of the transported material and that of the bed [23]. Often, this assumption is not valid in complex natural unsteady flows, such as, e.g., in an RoR reservoir where decelerated flow and sediment sorting occur [3,6]. Therefore, for a final judgment, it would be necessary to combine the strength of the experimental data, in particular GSDs from the colored sediments, with those of a numerical 3D simulation where the parameters determining sediment transport (e.g., bed shear stress) can be calculated with a high spatial and temporal resolution. The grain size distributions of the delta formation revealed that the sediments are finer on the delta top than on the delta tail. This striking result can only be explained if vertical sorting effects are considered. The use of colored sediments and the newly developed method (Section 2.3.3) proved to be well suited to study GSDs in combination with bed forms. It is also well suited to study vertical sorting effects in the future based on experimental results as well as on theoretical approaches [33,34].

Figure 9 illustrates a scheme of the delta formation together with water surface profiles as they evolve over time. The scheme was derived from the delta formation experiments in Section 3.2. The scheme applies for delta formations at the head of a reservoir of run-of-river hydropower plants in medium-sized gravel bed rivers where bed load transport is dominant. The delta formation consists of a steep delta front and a nearly horizontal delta top. In upstream direction, the delta top is connected with the delta tail. In Figure 9, the operation level of the RoR during turbine operation is illustrated. The head of the reservoir is located where the water depth h equals the water depth of the free-flowing section $h_0 + 1\%$, i.e., $h = 1.01 \times h_0$. At the beginning of the delta formation process, sediments from the upstream free-flowing section enter the reservoir at the head of the reservoir. From this starting point, the delta formation process takes place in both upstream and downstream directions. From the reservoir head, a sediment front develops that migrates in downstream direction where water depths gradually increase and bed shear stresses successively decrease. As the delta moves further into the reservoir, it grows in height and the delta front steepens. As the delta grows in height, the delta top becomes almost horizontal or even adversely sloped. As long as the shear stress on the delta top is too low to initiate sediment re-mobilization, the delta top increases in height due to the incoming sediments from upstream. During this process, the shear stress increases until it reaches the critical threshold. This is when the sediment layer close to the delta front is re-mobilized and slides over the steep delta front. During this period, the delta moves further into the reservoir. The remobilization process on the delta top stops when the shear stress falls below the critical threshold. This process repeats itself. The maximum slope $S_{rep}$ of the front is limited by the corresponding angle of repose of the sediment mixture. In addition, the speed of the delta decelerates in longitudinal direction and over time. Over time, the adjacent delta tail approaches the slope of the initial bed $S_0$. In [35], it is reported that the slope of the delta top (which is termed "delta topset") is about half of the original bed slope $S_0$. In [19], delta slopes are classified into 20%, 50% and 100% of the original bed slope $S_0$. Menné and Kiel (1959, cited in [19]) relate the delta slope to a shape factor calculated as the ratio of reservoir length to average reservoir width. For long and "thin" reservoirs (high shape factor) the delta slope becomes flatter. The shape factor of the RoR of this study is around 70. According to Menné and Kiel, the predicted delta slope is about 20% of the original slope. All these reports on the delta slope are not in conflict with the scheme of Figure 9. The determined delta slope is partly a question of the definition of the length of the delta top. As the slope of the delta tail gradually approaches the original bed slope, a location from the delta front can be found where the slope of the delta top is 20% or 50% of $S_0$. According to the experimental results, the water surface deviation from the operation level starts on the flat delta top. From there, the water level rise develops in upstream direction. It is therefore reasonable to distinguish this flat part from the rest of the delta formation. The proposed definition of the delta scheme (Figure 9) explicitly differentiates between delta tail and delta top.

In [36], the delta formation process of Lake Nasser is described. The reservoir has a length of 500 km and widths ranging from 300 m up to 9300 m. Sediment transport processes are dominated by

suspended load. Within 34 years of operation, a delta developed, having a length of about 150 km and a maximum height of 60 m above the original bed level. The bed material of the delta contains more clay and silt than the original bed. Despite the obvious differences in dimensions and sediment composition between Lake Nasser and the RoR reservoir of the present study, some similarities could be found: The delta top became flatter over time and is almost horizontal, while the delta front steepened in both cases. A similar delta development was observed for the Bhakra Dam in India, as reported in [20]

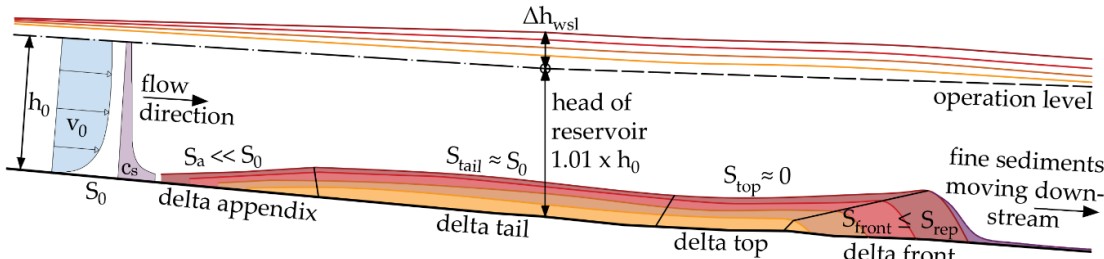

**Figure 9.** Spatio-temporal scheme of delta formation and corresponding water levels at the head of a run-of-river reservoir in medium-sized gravel bed rivers. Starting point is the bed slope $S_0$, i.e., the dynamic equilibrium slope of the free-flowing section and parallel water surface profiles which merge into the turbine operation level at the head of the reservoir. $S_{rep}$ = the slope corresponding to the angle of repose of the sediment mixture.

In the present study, the water levels in the reservoir are kept at operation level. Under these conditions, sediment transport at the reservoir head stops for grain sizes > 14 mm during a flow rate of $0.7 \times HQ_1$ at 1:1 scale. Almost all incoming sediments settle in the vicinity of the reservoir head leading to a substantial rise in water levels. The longer the flood event lasts, the further the delta formation extends in upstream direction. For a $0.7 \times HQ_1$ flood event, the water level rise will not increase the flood risk because the bank protection of the reservoir is designed to withstand much larger floods, e.g., a $HQ_{100}$. However, if for flow rates of high occurrence, the reservoir is not drawn down, delta and water levels will grow even more. This could endanger flood safety in case of high flood events. In addition, gravel fractions which accumulate in the delta formation might lead to a deficit of these ecologically important fractions further downstream. Sediment deficit can also lead to river bed incisions with severe technical, economic and ecological implications [37]. The $0.7 \times HQ_1$ was the flow rate separating the flat from the steep transport curve (Figure 5). The selected simplified flood event of high probability (Figure 2b) proved to be appropriate to mobilize a considerable amount of sediments (Table 4). Interestingly, the pre-calculated sediment transport capacities were found to be enveloping curves for the measured transport capacities (Figure 5). From a flood risk perspective and an ecological perspective, it is therefore reasonable to drawdown reservoirs at least at flow rates about $0.7 \times HQ_1$. This is particularly true if the dead storage capacity is filled with sediment and the bed has reached the gate elevation. Drawdown operations are then essential to prevent further deposition. During flushing events, the gate elevation, i.e., weir height, plays a key role to enhance sediment connectivity [3].

On the other hand, reservoir drawdowns are always accompanied by energy revenue losses and other costs. In particular, if the RoR is part of a chain of hydropower plants, drawdown and refill times might increase even further. The measured transport rates have a small gradient for flow rates $< 0.7 \times HQ1$ and a gradient about three times larger for flow rates $\geq 0.7 \times HQ1$. For flood events with a peak flow $< 0.7 \times HQ1$, the cost-benefit factor of reservoir drawdowns might not work out.

## 5. Conclusions

In this study, the head of a reservoir of a run-of-river hydropower plant (RoR) was examined in a physical model at a scale of 1:20. An idealized medium-sized gravel bed river of 20 m width and a slope of 0.5% was built based on the Muerz River in Styria, Austria. Colored sediments of five

different grain sizes were used, covering a range of 14–120 mm at 1:1 scale. Two main test series were performed. First, for a free-flowing section, sediment transport capacities and the conditions of dynamic equilibrium were determined for flow rates between $0.5 \times HQ_1$ and $0.8 \times HQ_1$. Second, for a flow rate of $0.7 \times HQ_1$, delta formation and sorting processes at the head of an RoR during turbine operation were investigated. A supportive analysis using different bed-load transport models was performed. For the free-flowing section, none of the formulas predicted the bimodal measured transport rates correctly, but most of the formulas predicted the measured transport rates with reasonable agreement. For the delta formation experiments, however, none of the models showed a good overall performance underlying the importance of further research on nonuniform sediment transport in complex turbulent flows. The grain size distributions of the delta formation suggest that vertical sorting effects play a dominant role. In the future, vertical sorting can be studied experimentally by using colored sediments and by applying the developed maximum likelihood classification method.

The delta formation experiments revealed that almost all incoming sediments accumulate at the head of the reservoir. Over time, the resulting delta formation grows in height and moves further into the reservoir. Four areas of the delta can be distinguished: (i) a steep delta front followed by an almost horizontal (ii) delta top which transitions into a (iii) delta tail. The delta tail has a substantial extent in upstream direction raising bed and water levels and approaching the slope of the free-flowing section. Upstream of the tail, a (iv) flat appendix merges into the original river bed.

The delta formation has the potential to increase the flood risk for high floods. It also has ecological impacts. Important spawning gravel fractions might accumulate on the delta and are missing further downstream. From this point of view, it might be recommended to drawdown the reservoir for a flow rate of $0.7 \times HQ_1$.

**Author Contributions:** Conceptualization, C.S.; methodology, C.S., T.G., K.R.; formal analysis, C.S., K.R., T.G.; investigation, K.R., T.G.; writing—original draft preparation, C.S.; writing—review and editing, T.G., K.R., H.H., C.H.; visualization, T.G., K.R., C.S. All authors have read and agreed on the published version of the manuscript.

**Funding:** The financial support by the Austrian Federal Ministry for Digital and Economic Affairs, the National Foundation of Research, Technology and Development of Austria is gratefully acknowledged. We gratefully acknowledge financial support from the Christian Doppler Research Association.

**Acknowledgments:** We kindly acknowledge the support of Johannes Schobesberger, Nora Lasinger and Elena Leutgoeb.

**Conflicts of Interest:** The authors declare no conflict of interest. The funders had no role in the design of the study; in the collection, analyses, or interpretation of data; in the writing of the manuscript, or in the decision to publish the results.

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
