# Peer review of "Experimental Study at the Reservoir Head of Run-of-River Hydropower Plants in Gravel Bed Rivers. Part I: Delta Formation at Operation Level"

_water, doi:10.3390/w12072035_

Round 1
Reviewer 1 Report
This article presents an experimental study on delta formation in run-of-river reservoirs.
The scaling approach used in this study is highly debatable, especially for sediment. Therefore, I don't think
that results can be quantitatively transferred to the 1:1 scale.
The experimental approach is not always clearly described.
Load calculations for the simplified flood scenario (figure 1b) are based on very strong assumptions that are not met.
More in depth interpretation of the results and comparison with literature or field data are needed.
The article is sometimes hard to read because of the lack of punctuation.

Reviewer 2 Report
This paper is well planned and constructed. The abstract and introduction are good. The rational for the paper and the experimental set up and methods are presented clearly. The results are also clear and well presented.
The delta progradation and grain size data are adequately covered in the discussion. However, I would have like to see a greater discussion of the practical implementation into sediment management. How effective would a flood of 0.7 HQ1 be in flushing the delta deposits through the reservoir and into the river below the radial gate? How often should this be done? What would the triggers for a flushing event be? Would it be related to the extent of the delta or rise in water level? Maybe this relates to my lack of knowledge about the operating conditions of a RoR hydropower plant, if so it would be useful to have the information about flushing the system discussed. This additional discussion would not need to be extensive but it would round out the paper and give more credence to the management implications in the title.
The paper has a few areas where the written expression could be improved slightly and I have suggested a few minor corrections in the text on the attached pdf file.
